# Deregulation of Secreted Frizzled-Related Protein 5 in Nonalcoholic Fatty Liver Disease Associated with Obesity

**DOI:** 10.3390/ijms22136895

**Published:** 2021-06-27

**Authors:** Laia Bertran, Marta Portillo-Carrasquer, Carmen Aguilar, José Antonio Porras, David Riesco, Salomé Martínez, Margarita Vives, Fàtima Sabench, Eva Gonzalez, Daniel Del Castillo, Cristóbal Richart, Teresa Auguet

**Affiliations:** 1Grup de Recerca GEMMAIR (AGAUR)-Medicina Aplicada (URV), Departament de Medicina i Cirurgia, University Rovira i Virgili (URV), Institutue d’Investigació Sanitària Pere Virgili (IISPV), 43007 Tarragona, Spain; laia.bertran@urv.cat (L.B.); marta.portillo.carrasquer@gmail.com (M.P.-C.); caguilar.hj23.ics@gencat.cat (C.A.); crichart.hj23.ics@gencat.cat (C.R.); 2Servei Medicina Interna, Hospital Universitari de Tarragona Joan XXIII, Mallafré Guasch, 4, 43007 Tarragona, Spain; aporras.hj23.ics@gencat.cat (J.A.P.); david_riesco@hotmail.com (D.R.); 3Servei Anatomia Patològica, Hospital Universitari de Tarragona Joan XXIII, Mallafré Guasch, 4, 43007 Tarragona, Spain; mgonzalez.hj23.ics@gencat.cat; 4Servei de Cirurgia, Hospital Sant Joan de Reus, Departament de Medicina i Cirurgia, Universitat Rovira i Virgili (URV), IISPV. Avinguda Doctor Josep Laporte, 2, 43204 Reus, Spain; mvives@gmail.com (M.V.); fatima.sabench@urv.cat (F.S.); danieldel.castillo@urv.cat (D.D.C.); 5Laboratori Clínic Institut Català de Salut (ICS), Hospital Universitari de Tarragona Joan XXIII, Mallafré Guasch, 4, 43007 Tarragona, Spain; egonzalezcarballo@hotmail.com

**Keywords:** SFRP5, nonalcoholic fatty liver disease, inflammation, liver, noncanonical WNT pathway

## Abstract

Secreted frizzled-related protein 5 (SFRP5), an antagonist of the noncanonical WNT pathway, has a controversial role in liver disease. The aim of this study was to analyze the role of SFRP5 and the noncanonical WNT pathway in nonalcoholic fatty liver disease (NAFLD). Plasma SFRP5 levels were determined by ELISA in women with normal weight (NW; *n* = 20) and morbid obesity (MO; *n* = 69). Women with MO were subclassified according to hepatic histology into normal liver (NL; *n* = 28), NAFLD (*n* = 41) (simple steatosis (SS; *n* = 24), and nonalcoholic steatohepatitis (NASH; *n* = 17)). We used RT-qPCR to evaluate the hepatic mRNA expression of SFRP5, WNT5A, and JNK in women with MO. SFRP5 levels were lower in NW than in MO patients who underwent a very low-calorie diet before surgery. Hepatic SFRP5 mRNA expression was higher in SS than in NL or NASH; additionally, patients with hepatic inflammation or ballooning presented lower SFRP5 abundance. WNT5A and JNK expression was enhanced in NAFLD compared with NL. In conclusion, circulating SFRP5 levels depend on the diet, and hepatic SFRP5 seems to have a protective role in the first steps of NAFLD; however, SFRP5 could be deregulated in an advanced stage while WNT5A and JNK are activated, promoting liver damage.

## 1. Introduction

Nonalcoholic fatty liver disease (NAFLD) is a chronic metabolic disorder of the liver that has emerged as a major public health concern with a high prevalence rate ranging from 12.9% to 46.0%, depending on the country [1]. NAFLD represents a continuum of liver abnormalities from simple steatosis (SS) to nonalcoholic steatohepatitis (NASH) that can lead to cirrhosis and liver cancer, emerging as an important cause of liver transplant [2,3,4]. The pathogenesis of NAFLD is complicated and involves environmental and genetic factors, hormone secretion, lipid peroxidation damage, immunological reactions, oxidative stress, abnormal fat metabolism, intestinal dysbiosis, and angiogenesis, which ultimately induce NASH and cirrhosis [5]. However, these mechanisms are not well understood, and effective methods for preventing and treating NAFLD are still lacking [5]. Therefore, improving the knowledge of the molecular pathways involved in NAFLD pathogenesis is crucial to identify future therapeutic targets.

Secreted frizzled-related protein 5 (SFRP5) is an anti-inflammatory adipocytokine member of the SFRP family [6,7] that is an antagonist of the WNT (wingless-MMTV integration site) family member 5a (WNT5A), a ligand of WNT pathway [8]. There are two types of WNT signaling pathways: canonical and noncanonical, but SFRP5 is implicated only in the last one. The noncanonical pathway is activated by WNT ligands that bind to the frizzled receptors, followed by the phosphorylation of Jun N-terminal kinase (JNK) that triggers proinflammatory and proliferative processes [8,9].

The SFRP5/WNT5A-mediated noncanonical pathways are associated with the pathogenesis of many inflammation-related diseases [10]. Although it has been reported that WNT signaling is activated during hepatic fibrosis, little is known about its precise mechanism in the liver [11]. Additionally, it is also known that the phosphorylation of JNK is implicated in cell death, cancer, T2DM, and obesity and plays an important role in the activation of liver fibrosis [12]. Therefore, it seems that SFRP5/WNT5A-mediated noncanonical may be involved in NAFLD pathogenesis (Figure 1).

In this sense, some studies have shown that SFRP5 can regulate metabolic disorders by improving hepatic lipid metabolism, inhibiting the growth of adipocytes [13,14], and alleviating hepatic steatosis [6,9]. Regarding animal models, previous studies reported that SFRP5 is a protective adipokine for glucose tolerance, adipose inflammation, hepatic steatosis, and fibrosis, and its anti-inflammatory role is perturbed in animal models of obesity with T2DM [14,15,16]. However, the role of SFRP5 in humans is not clear.

All these facts together suggest that the SFRP5 signaling pathway may be involved in NAFLD pathogenesis. Therefore, the main objective of this study was to investigate the specific role of SFRP5 in NAFLD: on the one hand, analyzing serum SFRP5 levels in women with normal weight (NW) and morbid obesity (MO) with and without NAFLD; and on the other hand, analyzing the relative mRNA hepatic abundance of SFRP5 in women with MO with different degrees of NAFLD. As an additional objective, we wanted to study in the same cohort of patients, mRNA hepatic abundance of WNT5A and JNK, two of the main genes implicated in the noncanonical WNT pathway, and their relationship with NAFLD.

## 2. Results

### 2.1. Baseline Characteristics of Subjects

The clinical characteristics and biochemical measurements of the population are shown in Table 1. We first classified our cohort depending on their body mass index (BMI) as NW (BMI < 25 kg/m^2^, *n* = 20) and MO (BMI > 40 kg/m^2^, *n* = 69). Then, those with MO were subclassified according to hepatic histology as normal liver (NL; *n* = 28), SS (*n* = 24), and NASH (*n* = 17), which were comparable in terms of weight, BMI, insulin, homeostatic model assessment method-insulin resistance (HOMA2-IR), glycosylated hemoglobin (HbA1c), cholesterol, high-density lipoprotein cholesterol (HDL-C), low-density lipoprotein cholesterol (LDL-C), TG, aspartate aminotransferase (AST), gamma-glutamyltransferase (GGT), systolic blood pressure (SBP), and diastolic blood pressure (DBP).

Biochemical analyses indicated that patients with NW had significantly lower weight, BMI, HbA1c, TG, and alanine aminotransferase (ALT) than NL, SS, and NASH groups; also, levels of insulin, GGT, and alkaline phosphatase (ALP) were lower in NW compared to SS; meanwhile, AST and GGT levels of NW subjects were lower compared to NASH patients. On the other hand, our NW subjects presented higher levels of HDL-C than NAFLD patients (NL, SS, and NASH) and enhanced levels of cholesterol than those with SS. In the MO cohort, we found higher levels of glucose, ALT, and ALP in SS subjects than in the NL group and increased levels of ALP in SS than in the NASH group. It is important to note that any of the patients of our cohort who presented T2DM neither need pharmacological treatment since it has been reported that treatments for diabetes alter SFRP5 levels [13].

### 2.2. Evaluation of Serum SFRP5 Levels According to BMI and Hepatic Histology

To achieve the main objective of this study, we evaluated serum SFRP5 levels in a cohort of women with NW and MO (NL, SS, and NASH). Our results showed significantly lower levels of SFRP5 in NW patients than those with MO, specifically in NL, SS, and NASH subjects, as shown in Figure 2. Unfortunately, we did not find significant differences in SFRP5 levels between NL, SS, and NASH groups.

### 2.3. Evaluation of Relative mRNA Abundance of SFRP5 in Liver According to Hepatic Histology

Apart from determining SFRP5 levels in our study cohort, to achieve the main objective of this study, we also evaluated the relative mRNA hepatic abundance of SFRP5 in a cohort of women with MO without or with NAFLD (SS and NASH). First, we classified our patients into NL and NAFLD, and we observed that the hepatic relative mRNA expression of SFRP5 was higher in the NAFLD group than in NL subjects (Figure 3A). Then, when we analyzed hepatic relative mRNA expression of SFRP5 according to different degrees of NAFLD, we found that SFRP5 expression was significantly higher in patients with SS than those with NL or NASH. However, hepatic relative mRNA abundance of SFRP5 did not show significant differences between NL and NASH subjects (Figure 3B).

### 2.4. Evaluation of the Relative mRNA Abundance of SFRP5 in Liver According to Liver Inflammation-Related Parameters

First, we wanted to explore the implication of SFRP5 in hepatic inflammation and ballooning processes analyzed under the microscope by an expert pathologist. In this regard, we observed decreased levels of mRNA abundance of SFRP5 in patients with liver inflammation than those without it (Figure 4A). We also found lower levels of mRNA relative abundance of SFRP5 in subjects with hepatic ballooning than those in the absence of it (Figure 4B). Finally, we observed low levels of hepatic mRNA SFRP5 abundance in patients with lobular inflammation than subjects without lobular inflammation (Figure 4C). There were no significant differences in SFRP5 mRNA hepatic relative expression according to portal inflammation presence or absence (Figure 4D).

### 2.5. Hepatic Expression of Validated Proinflammatory Molecules in SS and NASH Groups

Then, to corroborate the veracity of the expression peak of SFRP5 in SS samples and the downregulation in NASH ones, we have also analyzed the hepatic expression of IL-6 and TNF-α, well-known proinflammatory molecules, in SS and NASH groups. Therefore, we found significant differences in IL-6 and TNF-α expressions between SS and NASH (*p* = 0.033, *p* = 0.006, respectively), as expected.

### 2.6. Correlations of Relative Hepatic mRNA Abundance of SFRP5 with Clinical and Biochemical Parameters and with the Studied Adipocytokines

To deepen the knowledge of the role of SFRP5 in NAFLD pathogenesis, we wanted to analyze correlations between SFRP5 relative expression in the liver with different parameters such as weight, BMI, glucose, insulin, cholesterol, HDL-C, LDL-C, TG, and liver transaminases. In the correlation analysis, we only found a significant positive association between SFRP5 hepatic relative expression and ALP levels (*rho* = 0.404, *p* = 0.016).

### 2.7. Evaluation of Relative mRNA Abundance of WNT5A and JNK in Liver According to Hepatic Histology

To explore the implication of the WNT signaling pathway in NAFLD pathogenesis, we also wanted to analyze in our study cohort the hepatic mRNA abundance of WNT5A and JNK, two of the main genes involved in the WNT pathway together with SFRP5. On the one hand, we observed significantly higher mRNA relative expression of WNT5A in the liver of NAFLD patients than those with NL histology, as shown in Figure 5A. Moreover, when we analyzed the hepatic relative mRNA abundance of WNT5A according to different degrees of NAFLD, we found that WNT5A hepatic expression was significantly enhanced in patients with SS than those with NL or NASH (Figure 5B). On the other hand, we observed higher levels of hepatic JNK relative expression in NAFLD patients than NL subjects (Figure 5C). Additionally, we also found increased levels of hepatic mRNA abundance of JNK in SS patients than those with NL, as was graphically represented in Figure 5D.

### 2.8. Correlations of Relative Hepatic mRNA Abundance of WNT5A and JNK with Clinical, Biochemical, and Other Parameters

On the one hand, we found significant correlations between WNT5A relative hepatic expression and GGT. In addition, a positive correlation between WNT5A and SFRP5 relative hepatic expressions existed (Table 2).

On the other hand, when we analyzed correlations between JNK relative expression in liver and clinical and biochemical parameters and SFRP5 and WNT5A hepatic expressions, we found some associations, which are shown in Table 3.

### 2.9. Evaluation of Relative mRNA Abundance of SFRP5, WNT5A and JNK in Liver According to Hepatic Histology

Given that SFRP5, WNT5A, and JNK mRNA hepatic abundance have shown a differential expression in NAFLD compared to NL subjects, we wanted to compare these expressions among them, according to the histopathological classification of the liver, as was represented in Figure 6. In this sense, the relative mRNA abundance of SFRP5, WNT5A, and JNK of NL subjects showed similar expression levels. Nevertheless, in the SS group, we found increased mRNA expressions of SFRP5, WNT5A, and JNK than NL, but SFRP5 showed a higher increase compared to the other genes. In contrast, when we analyzed the NASH cohort, we observed that the mRNA expression was decreased again in SFRP5 and WNT5A, but this reduction was only significant in SFRP5.

## 3. Discussion

The main novelty of the present study is that we analyzed in a well-characterized cohort of women with MO and NAFLD the involvement of SFRP5 and the noncanonical WNT pathway in NAFLD pathogenesis. In this sense, although we did not find significant differences in serum SFRP5 levels between patients with or without NAFLD, we reported increased hepatic mRNA abundance of SFRP5 in patients with SS in contrast to those with NL or NASH. Moreover, we also found enhanced mRNA hepatic expressions of WNT5A and JNK, two of the main genes involved in the noncanonical WNT pathway, in SS compared to NL.

In the first instance, regarding obesity, we reported higher serum SFRP5 levels in patients with MO than NW subjects. These results are contrary to previous studies since Hu et al. showed that subjects with obesity had lower circulating SFRP5 levels compared to subjects with NW [13]. Moreover, Tan et al. demonstrated low SFRP5 levels in children with obesity [17]. This contradiction can be explained because our MO patients underwent a very low-calorie diet 3 months before the bariatric surgery, while our NW subjects followed a normal diet before the blood extraction. In this regard, the Tan et al. and Hu et al. cohorts of patients did not follow this type of strict diet. Additionally, some of the children with obesity in Tan et al.’s study underwent a reduction in caloric intake and reported an increase in SFRP5 levels after this lifestyle intervention [17], the same that occurred in our study. In addition, it should be taken into account that in both articles, they use male subjects; moreover, Tan et al.’s cohort is formed by children. Meanwhile, we studied adult women. This fact is also supported by a previous human study that demonstrated that SFRP5 can be used as a biomarker of the anti-inflammatory effect after caloric restriction [18]. Regarding our cohort with NAFLD, we did not find significant differences in SFRP5 serum levels between NL, SS, and NASH. Our results agree with Gutiérrez-Vidal et al., who reported non-significant differences between NAFLD groups neither between the first steps of fibrosis [19]. Regarding fibrosis, we could not perform this analysis because, in our study, any patient with NASH did not present hepatic fibrosis.

Later, we analyzed mRNA relative abundance of SFRP5 in the liver from MO patients with or without NAFLD. We reported increased expression of hepatic mRNA SFRP5 in NAFLD patients than NL ones; and then, when we classified our NAFLD subjects in SS and NASH, we observed higher expression in the SS group than NL and also than NASH patients. In this sense, there is only one previous human study in which they have analyzed hepatic mRNA expression of SFRP5 in NAFLD. They reported a decrease in SFRP5 expression in the liver as NAFLD progresses. However, when the data is deeply analyzed, this reduction was not significant between controls and SS subjects, but it was significant between controls and the NASH group [19]. These discrepancies can be given due to the fact that Gutierrez-Vidal et al. used the expression of actin-β to standardize the hepatic abundance of SFPR5 [19], while we used 18s expression, and also, they did not normalize the expression in relation to the control group. Moreover, given that inflammation and ballooning are two of the main finding of NASH [20], we wanted to explore the SFRP5 abundance differences according to the presence or absence of inflammation (portal and lobular) and hepatic ballooning. Our results showed that SFRP5 was lower in patients with general inflammation, lobular inflammation, and ballooning. These results can be explained by the fact that SFRP5 is an anti-inflammatory molecule that seems to have a protective role in the first steps of hepatic steatosis, but then the inflammation seems to deregulated SFRP5 signaling, blocking its inhibition of the noncanonical WNT pathway, which promotes NAFLD progression.

Since it was reported that the noncanonical WNT pathway involving JNK activation had been implicated in fatty liver disease [11,14], we wanted to study the hepatic relative mRNA expression of WNT5A and JNK to add new knowledge about the role of this pathway in NAFLD pathogenesis. On the one hand, we found higher relative mRNA WNT5A expression in patients with NAFLD than those with NL histology; concretely, we observed significantly enhanced expression in SS subjects than NL ones. On the other hand, we reported increased expression of JNK in NAFLD; in detail, we found higher levels in SS than in NL. However, we could not find significant differences in hepatic WNT5A and JNK expressions between SS and NASH patients. In this regard, our results support the fact that WNT5A and JNK are upregulated in NAFLD, triggering noncanonical WNT signaling that promotes liver damage [21], but we could not demonstrate that this pathway is increased in an advanced stage of NAFLD, such as NASH. In this regard, WNT5A and JNK hepatic expression analysis in NAFLD human subjects is a novelty, but our results seemed to agree with other studies assessed in animal models, such as Wang J. et al., who observed that the WNT pathway could play a key role in hepatic stellate cell activation and proliferation that trigger liver regeneration [11]. Moreover, Wang S. et al. postulated an association between the noncanonical WNT pathway and NAFLD, liver inflammation, and fibrosis [21]. Additionally, Kodama et al. indicated that blocking JNK may prevent the development of steatosis in mice models [22], which could represent a therapeutic target for this disease. Furthermore, Hirosumi et al. said that the activation of JNK was observed in the liver of obese mice, and Jnk1 knockout mice were protected from the development of obesity and insulin resistance [23].

At last, we observed a correlation between hepatic mRNA expression of WNT5A and GGT. In this sense, Coccia et al. demonstrated that GGT levels significantly increase with steatosis and fibrosis grade [24]. Moreover, we could demonstrate that hepatic mRNA expression of SFRP5, WNT5A, and JNK was correlated. This fact was explained because the three genes significantly increased in the steatosis state, but only SFRP5 expression significantly decreased in NASH, while WNT5A and JNK were maintained. Thus, it seems that in the first states of NAFLD, SFRP5 competes against the activation of the WNT pathway as a protective molecule. Then, in an advanced stage of liver damage and inflammation, its effects seem to be deregulated, and noncanonical WNT pathway signaling is activated, triggering NAFLD progression.

In this study, our cohort of women with MO has made it possible to establish some relationships between the WNT pathway (SFRP5, WNT5A, and JNK) and NAFLD without the interference of such confounding factors as sex or age. However, these results cannot be extrapolated to men, other obesity groups, or patients who do not follow a caloric restriction. Further studies, including these cohorts, would be useful in order to validate our findings.

## 4. Materials and Methods

### 4.1. Subjects

The institutional review board (Institut Investigació Sanitària Pere Virgili (IISPV) CEIm; 23c/2015; 11 May 2015) approved the study, and all participants provided written informed consent. The study population consisted of 89 Caucasian women: 20 with NW (BMI < 25 kg/m^2^) and 69 with MO (BMI > 40 kg/m^2^). Liver biopsies were obtained during planned laparoscopic bariatric surgery. All liver biopsies were indicated for clinical diagnosis. The exclusion criteria were as follows: (1) subjects who had alcohol consumption higher than 10 g/day; (2) patients who had acute or chronic hepatic, inflammatory, infectious, or neoplasic diseases; (3) women who were menopausal or undergoing contraceptive treatment; (4) women with diabetes receiving pioglitazone or insulin; and (5) patients treated with antibiotics in the previous 4 weeks.

### 4.2. Sample Size

Accepting an α risk of 0.05 and a β risk of less than 0.2 in a bilateral contrast, 24 subjects per group were needed to detect a difference ≥ 0.2 units. It is assumed that the common standard deviation is 0.3.

### 4.3. Liver Pathology

Liver samples were stained with hematoxylin and eosin and Manson’s trichrome stains and scored by experienced hepatopathologists using the methods described elsewhere [25]. Hepatic samples were evaluated depending on the presence and the degree of macrovesicular steatosis to diagnose simple steatosis, and also portal and lobular inflammation, fibrosis (intensity and location), hepatocellular ballooning, and Mallory’s hyaline to identify NASH samples. Samples with less than 5% of macrovesicular steatosis without the presence of inflammation were defined as the normal liver. According to their liver pathology, women with MO were classified into NL histology (*n* = 28) and NAFLD (*n* = 41). Then, patients were subclassified into NL histology (*n* = 28), SS (micro/macrovesicular steatosis without inflammation or fibrosis, *n* = 24), and NASH (Brunt grades 1–2, *n* = 17). None of the patients with NASH in our cohort presented fibrosis or cirrhosis.

### 4.4. Biochemical Analyses

All of the subjects included underwent physical, anthropometric, and biochemical assessments. Blood extraction was performed by specialized nurses through a BD Vacutainer^®^ system after overnight fasting before bariatric surgery. Venous blood samples were obtained using empty and ethylenediaminetetraacetic acid tubes, which were separated into plasma and serum aliquots by centrifugation (3500 rpm, 4 °C, 15 min). Biochemical parameters were analyzed using a conventional automated analyzer. Insulin resistance was estimated using HOMA2-IR.

Peripheral SFRP5 levels were analyzed by enzyme-linked immunosorbent assay (ELISA) according to the manufacturer’s instructions (Ref. BA E-8900, Labor Diagnostika Nord, Nordhorn, Germany).

### 4.5. Gene Expression in the Liver

Liver samples collected during bariatric surgery were conserved in RNAlater (Qiagen, Hilden, Germany) at 4 °C and then processed and stored at −80 °C. Total RNA was extracted from the tissue by using the RNeasy mini kit (Qiagen, Barcelona, Spain). Reverse transcription to cDNA was performed with the High-Capacity RNA-to-cDNA Kit (Applied Biosystems, Madrid, Spain). Real-time quantitative PCR was performed with the TaqMan Assay predesigned by Applied Biosystems for the detection of SFRP5 (Hs00169366_m1), JNK (Hs01548508_m1), WNT5A (Hs00998537_m1), IL-6 (Hs00174131_m1), and TNF-α (Hs02621508_s1). The expression of each gene was calculated relative to the expression of 18s RNA (Fn04646250_s1), and then it was normalized using the control group (NL) as a reference. All reactions were carried in duplicate in 96-well plates using the 7900HT Fast Real-Time PCR system (Applied Biosystem, Foster City, CA, USA).

### 4.6. Statistical Analysis

The data were analyzed using the SPSS/PC+ for Windows statistical package (version 23.0; SPSS, Chicago, IL, USA). The Kolmogorov–Smirnov test was used to assess the distribution of variables. Continuous variables were reported as the mean (SD); non-continuous variables were reported as the median and the interquartile range. The different comparative analyses were performed using a nonparametric Mann–Whitney U test or Kruskal–Wallis test, according to the presence of two or more groups. The strength of the association between variables was calculated using Spearman’s method. *p*-values < 0.05 were statistically significant.

## 5. Conclusions

Circulating SFRP5 levels increase after caloric restriction in subjects with MO regardless of NAFLD status. However, hepatic SFRP5 could have a protective role in the first steps of NAFLD in an attempt to inhibit the noncanonical WNT pathway but could be deregulated at the advanced stage of the disease while WNT5A and JNK are activated, thus promoting liver damage.

## Figures and Tables

**Figure 1 ijms-22-06895-f001:**
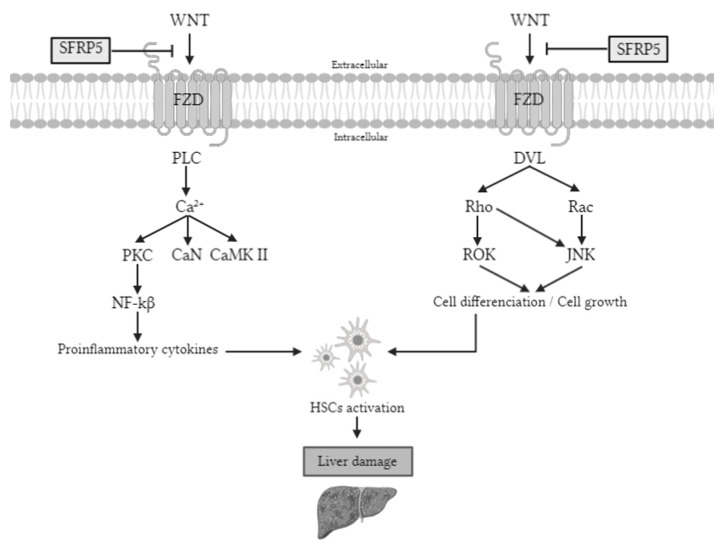
Noncanonical WNT pathway signaling. The WNT signaling pathway activates proteins that promote the synthesis of proinflammatory cytokines and cell differentiation. These two processes cause the activation of HSCs, which will cause liver damage if the stimulation persists. However, SFRP5 is able to inhibit the WNT signaling pathway, so the pathway is inactivated, so the liver is protected. PLC, phospholipase C; Ca^2+^, calcium ^2+^; PKC, protein kinase C; CaN, serine/threonine-protein phosphatase; CaMK II, Ca^2+^/calmodulin-dependent protein kinase II; NF-kβ, nuclear factor kappa-light-chain-enhancer of activated B cells; DVL, disheveled; ROK, rho-associated protein kinase; JNK, Jun N-terminal kinase; Wnt, wingless-MMTV integration site; Sfrp5, secreted frizzled-related protein 5; FZD, frizzled receptor; HSC, hepatic stellate cells.

**Figure 2 ijms-22-06895-f002:**
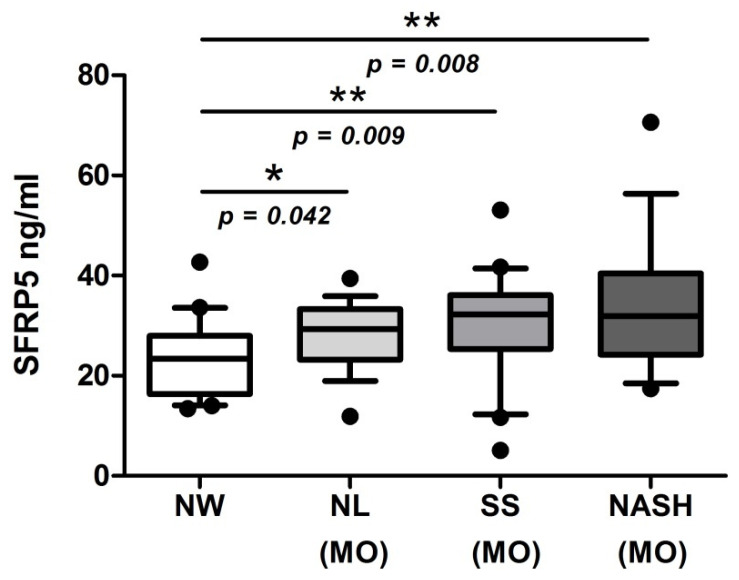
Box plot graphical representation of serum levels of SFRP5 women with NW and women with MO subclassified according to NL, SS, and NASH. NW, normal weight; MO, morbidly obesity; NL, normal liver; SS, simple steatosis; NASH, nonalcoholic steatohepatitis; SFRP5, secreted frizzled-related protein 5. Individual data points outside the maximum and minimum limits represent outliers. *p* < 0.05 was considered statistically significant (* means *p* < 0.05; ** means *p* < 0.01).

**Figure 3 ijms-22-06895-f003:**
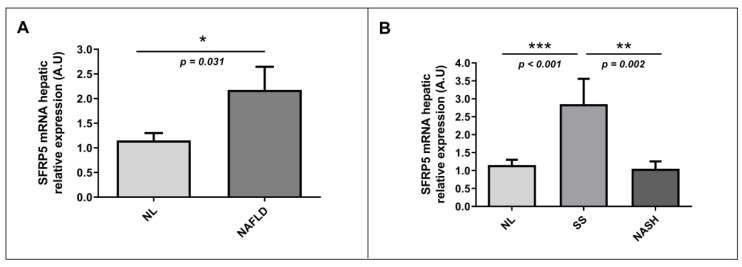
Differential relative mRNA abundance of SFRP5 in liver between (**A**) women with MO with NL histology and women with MO with NAFLD and (**B**) MO women with NL, SS, and NASH. A.U, arbitrary units; MO, morbid obesity; NAFLD, nonalcoholic fatty liver disease; NL, normal liver; SS, simple steatosis; NASH, nonalcoholic steatohepatitis; SFRP5; secreted frizzled-related protein 5. *p* < 0.05 was considered statistically significant (* means *p* < 0.05; ** means *p* < 0.01; *** means *p* < 0.001).

**Figure 4 ijms-22-06895-f004:**
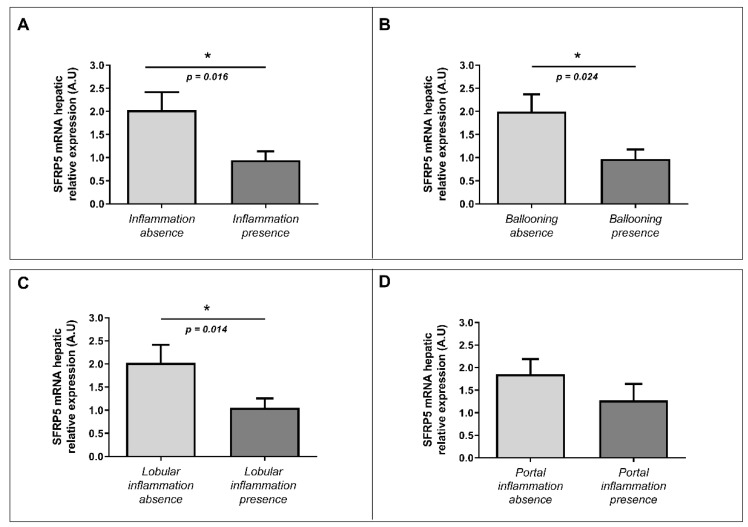
Differential relative mRNA hepatic expression of SFRP5 between different parameters related to inflammation (ballooning, portal inflammation, and lobular inflammation). Differential relative mRNA abundance of SFRP5 in liver between (**A**) absence of inflammation and inflammation presence; (**B**) ballooning absence and ballooning presence; (**C**) lobular inflammation absence and presence; and (**D**) portal inflammation absence and presence. A.U, arbitrary units; SFRP5; secreted frizzled-related protein 5. *p* < 0.05 was considered statistically significant (* means *p* < 0.05).

**Figure 5 ijms-22-06895-f005:**
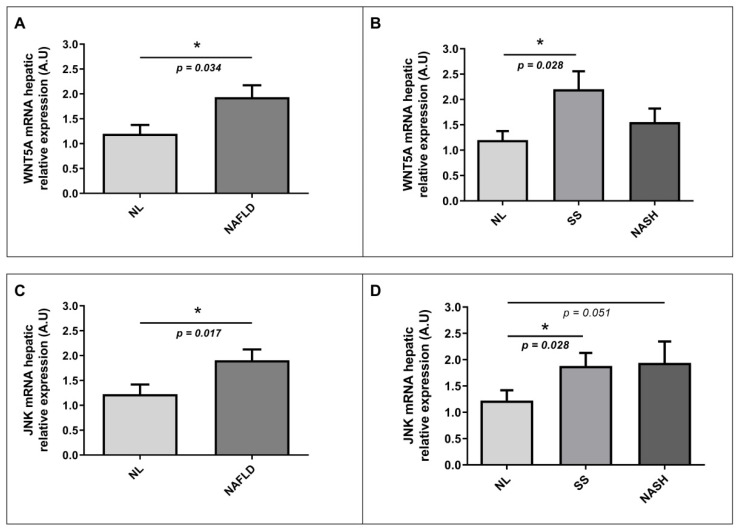
Differential relative mRNA abundance of WNT5A in liver between (**A**) women with MO and NL histology and women with MO with NAFLD; (**B**) women with MO with NL histology, women with MO with SS, and women with MO with NASH. Differential relative mRNA abundance of JNK in liver between (**C**) women with MO with NL histology and women with MO with NAFLD; (**D**) women with MO with NL histology, women with MO with SS, and women with MO with NASH. A.U, arbitrary units; MO, morbidly obesity; NAFLD, nonalcoholic fatty liver disease; NL, normal liver; SS, simple steatosis; NASH, nonalcoholic steatohepatitis; WNT5A, WNT family member 5a; JNK, Jun N-terminal kinase. *p* < 0.05 was considered statistically significant (* means *p* < 0.05).

**Figure 6 ijms-22-06895-f006:**
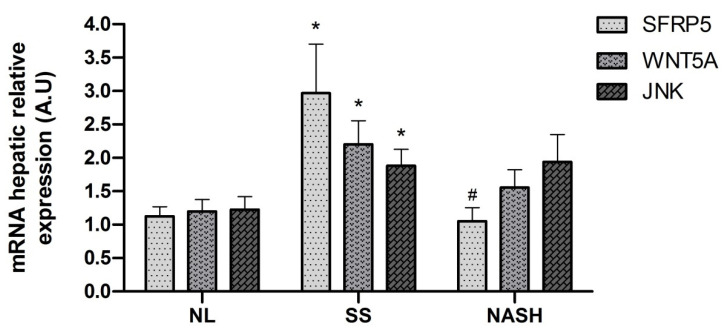
Differential relative mRNA abundance of SFRP5, WNT5A, and JNK in the liver. Differential relative mRNA abundance of SFRP5, WNT5A, and JNK in the liver of women with MO subclassified by NL, SS, and NASH. MO, morbid obesity; NL, normal liver; SS, simple steatosis; NASH, nonalcoholic steatohepatitis; SFRP5, secreted frizzled-related protein 5; WNT5A, WNT family member 5a; JNK, Jun N-terminal kinase; A.U, arbitrary units. * Significant differences between the NL group and the SS group (*p* < 0.05). # Significant differences between SS group and NASH group (*p* < 0.05).

**Table 1 ijms-22-06895-t001:** Anthropometric and biochemical variables of women in the studied cohort.

		MO (*n* = 69)		
	NW	NL	SS	NASH
Variables	(*n* = 20)	(*n* = 28)	(*n* = 24)	(*n* = 17)
Weight (kg)	54.20(51.00–64.25)	116.50(107.25–130.50) *	113.20(108.33–128.00) ^&^	112.00(104.65–125.00) ^^^
BMI (kg/m^2^)	21.97(20.47–23.90)	43.30(40.94–46.47) *	44.35(40.82–46.83) ^&^	44.46(40.76–46.03) ^^^
Glucose (mg/dL)	90.00(84.00–97.00)	85.50(76.25–93.00)	93.50(85.75–107.00) ^$^	91.00(82.50–101.20)
Insulin (mUI/L)	7.80(4.80–9.90)	9.43(5.59–16.21)	11.27(7.81–14.51) ^&^	6.57(5.09–23.04)
HOMA 2-IR	1.05(0.60–1.30)	1.23(0.75–2.05)	1.49(0.95–2.18)	0.86(0.61–3.00)
HbA1c (%)	4.75(4.50–5.03)	5.40(5.30–5.70) *	5.60(5.25–6.03) ^&^	5.50(5.20–6.10) ^^^
Cholesterol (mg/dL)	193.98 ± 30.66	171.88 ± 36.20	174.42 ± 35.41 ^&^	185.28 ± 43.39
HDL-C (mg/dL)	63.75 ± 16.03	40.84 ± 9.89 *	42.56 ± 12.38 ^&^	38.89 ± 8.47 ^^^
LDL-C (mg/dL)	109.99 ± 30.67	108.16±27.94	104.39 ± 31.21	104.62 ± 31.58
TG (mg/dL)	79.50(49.50–149.25)	106.00(89.00–136.00) *	129.50(85.75––175.50) ^&^	140.00(106.00–247.00) ^^^
AST (UI/L)	19.50(15.75–23.00)	20.50(15.75–36.25)	23.00(17.00–35.00)	24.00(17.00–43.00) ^^^
ALT (UI/L)	15.00(12.00–21.00)	22.00(16.00–27.00) *	31.00(23.00–35.75) ^$,&^	30.00(15.50–40.00) ^^^
GGT (UI/L)	12.00(9.00–20.00)	18.00(16.00–27.00)	21.00(16.25–30.50) ^&^	25.00(15.00–27.00) ^^^
ALP (Ul/L)	54.44 ± 14.10	60.42 ± 13.09	75.80 ± 11.66 ^$,&^	62.77 ± 11.16 ”
SBP (mmHg)	118.56 ± 10.92	119.00 ± 18.26	120.09 ± 13.41	113.44 ± 13.96
DBP (mmHg)	72.00(68.50–75.00)	63.00(57.75–75.75)	62.00(59.00–72.50)	65.50(56.75–70.75)

MO, morbid obesity; NW, normal weight; NL, normal liver; SS, simple steatosis; NASH, nonalcoholic steatohepatitis; BMI, body mass index; HOMA2-IR, homeostatic model assessment method-insulin resistance; HbA1c, glycosylated hemoglobin; HDL-C, high-density lipoprotein cholesterol; LDL-C, low-density lipoprotein cholesterol; TG, triglycerides; AST, aspartate aminotransferase; ALT, alanine aminotransferase; GGT, gamma-glutamyltransferase; ALP, alkaline phosphatase; SBP, systolic blood pressure; DBP, diastolic blood pressure. Data are expressed as the mean ± standard deviation or median (interquartile range) depends on the distribution of the variables. * Significant differences between the NW group and the NL group (*p* < 0.05). ^&^ Significant differences between the NW group and the SS group (*p* < 0.05). ^^^ Significant differences between the NW group and the NASH group (*p* < 0.05). ^$^ Significant differences between the NL group and the SS group (*p* < 0.05). ” Significant differences between the SS group and the NASH group (*p* < 0.05).

**Table 2 ijms-22-06895-t002:** Significant correlations between WNT5A relative hepatic expression with clinical and biochemical parameters and with SFRP5 and JNK hepatic expressions.

Variables	WNT5A mRNA Hepatic R.E.	
	*rho*	*p*-Value
GGT (Ul/L)	0.318	0.033
SFRP5 hepatic R.E.	0.535	<0.001
JNK hepatic R.E.	0.846	<0.001

SFRP5, secreted frizzled-related protein 5; JNK, Jun N-terminal kinase; GGT, gamma-glutamyltransferase; R.E., relative expression. Relative expression was calculated with 18s RNA as a housekeeping gene. Data are expressed as the correlation coefficient *rho* of Spearman and *p*-value (*p* < 0.05 was considered statistically significant).

**Table 3 ijms-22-06895-t003:** Significant correlations between JNK relative hepatic expression with clinical and biochemical parameters and with SFRP5 and JNK hepatic expressions.

Variables	JNK mRNA Hepatic R.E.	
	*rho*	*p*-Value
SFRP5 hepatic R.E.	0.513	0.001
WNT5A hepatic R.E.	0.846	<0.001

SFRP5, secreted frizzled-related protein 5; WNT5A, WNT family member 5a; R.E., relative expression. Relative expression was calculated with 18s RNA as a housekeeping gene. Data are expressed as the correlation coefficient *rho* of Spearman and *p*-value (*p* < 0.05 was considered statistically significant).

## Data Availability

Not applicable.

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
