# Peer review of "Deregulation of Secreted Frizzled-Related Protein 5 in Nonalcoholic Fatty Liver Disease Associated with Obesity"

_ijms, 2021, doi:10.3390/ijms22136895_

Round 1

Reviewer 1 Report

Why were only female subjects included in the study?

Did authors look at other validated markers for SS and NASH in their samples to lend more credibility to the unusually higher SFRP5 levels seen in SS?

Table 2, 3: Include that relative expression was calculated wrt 18s RNA.

Was normality of the data checked before Spearman correlation analysis? Given the small sample size, Pearson's correlation would have been more suitable.

Could differences in the sample processing and stability explain reduced levels of SFRP5 in NASH?

Page 9 and 13: Authors highlight the differences between published studies but should discuss more on how those may explain the contradictory outcomes.

Line 361: Change to 18s RNA

Reviewer 2 Report

The authors collected serum and liver biopsy samples from bariatric surgery patients and study the SFRP5 and JNK, and WNT5A expression. There are only some issues to be resolved.

  1. Table 1 shown * for significant differences between the NW group and NL group. The symbol should be in the NL, not in the NW.
  2. The Figure 2 protein level not consistent with the Figure 3B mRNA level of SFRP5. How do you explain that?
  3. Too many paragraphs to be integrated.
  4. The IRB numbers need to provide.
  5. The SS and NASH identification should be described in more detail for the definition.
